# Endogenous VIP VPAC_1_ Receptor Activation Modulates Hippocampal Theta Burst Induced LTP: Transduction Pathways and GABAergic Mechanisms

**DOI:** 10.3390/biology11050627

**Published:** 2022-04-20

**Authors:** Ana Caulino-Rocha, Nádia Carolina Rodrigues, Joaquim Alexandre Ribeiro, Diana Cunha-Reis

**Affiliations:** 1Departamento de Química e Bioquímica Faculdade de Ciências, Universidade de Lisboa, Campo Grande, 1749-016 Lisboa, Portugal; avrocha@fc.ul.pt; 2BioISI—Instituto de Biosistemas e Ciências Integrativas, Faculdade de Ciências, Universidade de Lisboa, Campo Grande, 1749-016 Lisboa, Portugal; 3Unidade de Neurociências, Instituto de Medicina Molecular, Faculdade de Medicina, Universidade de Lisboa, Av. Prof. Egas Moniz, 1649-028 Lisboa, Portugal; ncar21@gmail.com (N.C.R.); jaribeiro@fm.ul.pt (J.A.R.); 4Instituto de Farmacologia e Neurociências, Faculdade de Medicina, Universidade de Lisboa, Av. Prof. Egas Moniz, 1649-028 Lisboa, Portugal

**Keywords:** VIP, LTP, VPAC_1_ receptors, VPAC_2_ receptors, interneurons, Kv4.2, hippocampus

## Abstract

**Simple Summary:**

Regulation of synaptic plasticity through control of disinhibition is an important process in the prevention of excessive plasticity in both physiological and pathological conditions. Interneuron-selective interneurons, such as the ones expressing VIP in the hippocampus, may play a crucial role in this process. In this paper we showed that endogenous activation of VPAC_1_—not VPAC_2_ receptors—exerts an inhibitory control of long-term potentiation (LTP) induced by theta-burst stimulation (TBS) in the hippocampus, through a mechanism dependent on GABAergic transmission. This suggests that VPAC_1_-mediated modulation of synaptic transmission at GABAergic synapses to interneurons will ultimately influence NMDA-dependent LTP expression by modulating inhibitory control of pyramidal cell dendrites and postsynaptic depolarization during LTP induction. Accordingly, the transduction pathways mostly involved in this effect were the ones involved in TBS-induced LTP expression like NMDA receptor activation and CaMKII activity. In addition, the actions of endogenous VIP through VPAC_1_ receptors may indirectly influence the control of dendritic excitability by Kv4.2 channels.

**Abstract:**

Vasoactive intestinal peptide (VIP), acting on both VPAC_1_ and VPAC_2_ receptors, is a key modulator of hippocampal synaptic transmission, pyramidal cell excitability and long-term depression (LTD), exerting its effects partly through modulation GABAergic disinhibitory circuits. Yet, the role of endogenous VIP and its receptors in modulation of hippocampal LTP and the involvement of disinhibition in this modulation have scarcely been investigated. We studied the modulation of CA1 LTP induced by TBS via endogenous VIP release in hippocampal slices from young-adult Wistar rats using selective VPAC_1_ and VPAC_2_ receptor antagonists, evaluating its consequence for the phosphorylation of CamKII, GluA1 AMPA receptor subunits and Kv4.2 potassium channels in total hippocampal membranes obtained from TBS stimulated slices. Endogenous VIP, acting on VPAC_1_ (but not VPAC_2_) receptors, inhibited CA1 hippocampal LTP induced by TBS in young adult Wistar rats and this effect was dependent on GABAergic transmission and relied on the integrity of NMDA and CaMKII-dependent LTP expression mechanisms but not on PKA and PKC activity. Furthermore, it regulated the autophosphorylation of CaMKII and the expression and Ser_438_ phosphorylation of Kv4.2 potassium channels responsible for the A-current while inhibiting phosphorylation of Kv4.2 on Thr_607_. Altogether, this suggests that endogenous VIP controls the expression of hippocampal CA1 LTP by regulating disinhibition through activation of VPAC_1_ receptors in interneurons. This may impact the autophosphorylation of CaMKII during LTP, as well as the expression and phosphorylation of Kv4.2 K^+^ channels at hippocampal pyramidal cell dendrites.

## 1. Introduction

Vasoactive intestinal peptide (VIP) is an endogenous modulator of hippocampal synaptic transmission and excitability [1]. The actions of VIP in the hippocampus involve both VPAC_1_ and VPAC_2_ G protein-coupled receptors of the VIP/PACAP family. These receptors bind pituitary adenylate cyclase-activating polypeptide (PACAP) with similar affinity. A third receptor in this family, the PAC_1_ receptor, binds preferentially PACAP over VIP, which is also able to modulate hippocampal synaptic transmission [2].

Given that VIP is exclusively expressed by hippocampal interneurons [3], it is not surprising that its actions on synaptic transmission are mainly dependent on GABAergic transmission [1,4]. Three distinct populations of hippocampal interneurons express VIP, yet they selectively target different hippocampal cells. Thus, it is to be expected that VIP plays several fundamentally different roles in the control of hippocampal synaptic transmission. Not surprisingly, endogenous VIP has opposing actions on GABA release when activating VPAC_1_ or VPAC_2_ receptors [5]. Furthermore, VIP enhances synaptic transmission to CA1 pyramidal cells through inhibition of GABAergic interneurons that control pyramidal cell dendrites, leading to disinhibition [1]. This action is mediated mainly by VPAC_1_ receptors present at the *strata*
*oriens* and *radiatum* [1,6,7]. Still, VIP also enhances hippocampal pyramidal cell excitability independently of GABAergic transmission [1,8], through inhibition of the slow Ca^2+^-activated K^+^ current (IsAHP) [9], and likely by activating pyramidal cell layer VPAC_2_ receptors [1].

VIP is also paramount for many hippocampal-dependent learning processes [10,11,12,13,14] and, in effect, VIP deficient mice do not develop a number of hippocampal-dependent learning abilities, such as reversal learning [15]. Synaptic plasticity phenomena like long-term potentiation (LTP), long-term depression (LTD) and depotentiation are believed to underlie distinct aspects of these hippocampal-dependent processes, yet LTP and LTD are similarly dependent on NMDA receptor activity. We recently described that endogenous VIP, by activating VPAC_1_ receptors, is an critical endogenous modulator of NMDA-dependent LTD and depotentiation in hippocampal CA1 [16], synaptic plasticity subtypes associated with memory reversal, reformulation and stabilisation. Yang et al. also demonstrated an enhancement of NMDA currents in CA1 pyramidal cells by exogenously applied VIP, an effect mimicked by VPAC_2_ but not so markedly by VPAC_1_ selective agonists [17], further co-substantiating the role of these receptors in modulation of NMDA-dependent hippocampal synaptic plasticity phenomena. Even though neuropeptides are usually released by repetitive firing like the one occurring during TBS stimulation, no study, to date, has elucidated the role of endogenous VIP and its receptors to the modulation of hippocampal LTP, a synaptic plasticity subtype associated with hippocampal-dependent spatiotemporal memory acquisition, such as that occurring during spatial exploration [18]. Given the importance of VIP in the modulation of hippocampal-dependent learning and memory, this should also be investigated.

LTP can be evoked in vitro by theta burst stimulation (TBS) [19,20], a sequence of electrical stimuli that mimic CA1 pyramidal cell burst firing such as that which occurs during the hippocampal theta rhythm (4–10 Hz), an EEG pattern linked to hippocampal memory storage in rodents [21,22]. Bursts repeated at the theta frequency produce maximal LTP by suppressing feedforward inhibition with a first burst or a priming single pulse that allows for sufficient depolarization to activate NMDA receptors [23,24]. This process involves activation of GABA_B_ autoreceptors that strongly inhibit GABA release from feedforward interneurons [25]. Still, additional GABAergic mechanisms may play a role in TBS-induced LTP [26].

In this paper, we investigated the role of endogenous VIP in the modulation of hippocampal LTP induced in vitro by mild TBS, the receptors and transduction pathways operated by VIP in this modulation and its dependency on GABAergic transmission. A preliminary account of some of the results has been published as an abstract.

## 2. Methods

All protocols and procedures were performed according to ARRIVE guidelines for experimental design, analysis, and their reporting. Animal housing and handing was performed in accordance with the Portuguese law (DL 113/2013) and European Community guidelines (86/609/EEC and 63/2010/CE). The experiments were conducted on hippocampal slices from young-adult (6–7 weeks old) male outbred Wistar rats (Harlan Iberica, Barcelona, Spain) as previously described [18]. Female rats were not used due to hormonal influences on LTP. Animals were first anesthetized with halothane and decapitated. The hippocampus was immediately dissected free in ice-cold artificial cerebrospinal fluid (aCSF), composition in mM: NaCl 124, KCl 3, NaH_2_PO_4_ 1.25, NaHCO_3_ 26, MgSO_4_ 1, CaCl_2_ 2, glucose 10, and gassed with a 95% O_2_–5% CO_2_ mixture.

### 2.1. LTP Experiments

Electrophysiological recordings were made in hippocampal slices (400 µm thick) cut perpendicularly to the long axis of the hippocampus using a McIlwain tissue chopper essentially as described [18]. Slices were kept in a resting chamber in the same gassed aCSF at room temperature 22–25 °C for at least 1 h to allow their energetic and functional recovery, then one slice at a time was transferred to a submerged recording chamber of 1 mL capacity, where it was continuously superfused at a rate of 3 mL/min with the same gassed solution at 30.5 °C. Stimulation (rectangular pulses of 0.1 ms) was delivered through a bipolar concentric wire electrode placed on the Schaffer collateral/commissural fibres in the *stratum radiatum*, located approximately at mid distance from the hippocampal fissure and the pyramidal cell layer. Two separate sets of the Schaffer pathway (S1 and S2) were stimulated alternately every 10 s, each pathway being stimulated every 20 s (0.05 Hz). (Figure 1A). Field excitatory post-synaptic potentials (fEPSP, Figure 1A) were recorded extracellularly from CA1 *stratum radiatum* using micropipettes filled with 4 M NaCl and of 2–4 MΩ resistance. The fEPSPs were evoked on the two pathways and the initial intensity of the stimulus was that eliciting a fEPSP of 600–1000 mV amplitude (about 50% of the maximal response), while avoiding signal contamination by the population spike, and of similar magnitude in both pathways. This is in the perfect dynamic range to induce both LTP and LTD/depotentiation. The averages of six consecutive responses from each pathway were obtained, measured, graphically plotted and recorded for further analysis with a personal computer using the LTP software [27]. The fEPSPs were quantified as the slope of the initial phase of the potential.

The independence of the two pathways was tested at the end of the experiments by studying paired-pulse facilitation (PPF) across both pathways, less than 10% facilitation being usually observed. To elicit PPF, the two Schaffer pathways were stimulated with 50 ms interpulse interval. The synaptic facilitation was quantified as the ratio P2/P1 between the slopes of the fEPSP elicited by the second P2 and the first P1 stimuli.

LTP was induced by a mild TBS pattern (five trains of 100 Hz, 4 stimuli, separated by 200 ms). The stimulation protocol used to induce LTP was delivered once a stable baseline was observed for at least 20 min. Stimulus intensity was not altered during these stimulation protocols. LTP intensity was calculated as the % change in the average slope of the potentials taken from 50 to 60 min after the induction protocol, compared to the average slope of the fEPSP measured during the 10 min that preceded the induction protocol. Data normalisation was used to control for unwanted sources of variation. Control and test conditions were tested in independent pathways in the same slice. In all experiments, S1 always refers to the first pathway (left or right, randomly assigned) to which TBS was applied. Test drugs were added to the perfusion solution 20 min before TBS delivery to the test pathway (S2) and were present until the end of the experiment except for the anti-VIP antibody that was present only until 20 min after TBS. When testing the effect of PG 97-269 in the presence of other drugs (bicuculine, SKF89976a, AP-5, GF109203x, H-89 or KN-62), these were added to the perfusion media at least 30 min before TBS stimulation of the control pathway (S1) and were present until the end of the experiment. Each *n* represents a single LTP experiment (S1 vs. S2 conditions) performed in one slice from an independent animal, i.e., *n* denotes the number of animals.

### 2.2. Western Blot Analysis of CamKII, GluA1 and Kv4.2 Phosphorylation

For phosphorylation analysis, hippocampal slices were obtained, as described above, and allowed to recover. Slices were then brought to the electrophysiology chamber and superfused at a flow rate of 3 mL/min with gassed aCSF at 30.5 °C. Stimulation was delivered every 15 s in the form of rectangular pulses (0.1 ms duration) using a bipolar concentric wire electrode laid on the Schaffer collateral/commissural fibres in the *stratum*
*radiatum* and continued for 80 min (average duration of an electrophysiological experiment). In test (but not in control) slices TBS was delivered 20 min after starting basal stimulation that continued after TBS and until the end of the experiment. These conditions were tested either in the presence or the absence of the VPAC_1_ antagonist PG 97-269 (100 nM). The occurrence of an LTP was confirmed using electrophysiological recordings essentially as described above. Total hippocampal membranes for Western blot studies were obtained from these slices as previously described [26]. Briefly, at the end of stimulation, slices were collected in sucrose solution (320 mM Sucrose, 1 mg/mL BSA, 10 mM HEPES e 1 mM EDTA, pH 7,4) containing protease (complete, mini, EDTA-free Protease Inhibitor Cocktail, Sigma) and phosphatase (1 mM PMSF, 2 mM Na_3_VO_4_, and 10 mM NaF) inhibitors and homogenized with a Potter–Elvejham apparatus. Each sample (*n* = 1) was obtained from several slices (minimum 4 per condition) from 2–3 animals. The suspension was centrifuged at 1500× *g* for 10 min and the supernatant collected and further centrifuged at 14,000× *g* for 12 min. The pellet was washed twice with modified aCSF (20 mM HEPES, 1 mM MgCl_2_, 1.2 mM NaH_2_PO_4_, 2.7 mM NaCl; 3 mM KCl, 1.2 mM CaCl_2_, 10 mM glucose, pH 7.4) also containing protease and phosphatase inhibitors and hippocampal membranes were resuspended at a concentration of 1 mg/mL protein concentration (Bradford assay) in modified aCSF. This suspension of hippocampal membranes was snap-frozen in liquid nitrogen and stored in aliquots at −80 °C until Western blot analysis. All samples were analysed in duplicate in Western blot experiments, except for CaMKII autophosphorylation.

For Western blot, samples processed as above were incubated for 5 min at 95 °C with Laemmli buffer (125 mM Tris-BASE, 4% SDS, 50% glycerol, 0.02% Bromophenol Blue, 10% β-mercaptoethanol), run on standard 10% sodium dodecyl sulphate polyacrylamide gel electrophoresis (SDS-PAGE) and transferred to PVDF membranes (Immobilon-P transfer membrane PVDF, pore size 0.45 μm, Immobilon). After blocking for 1 h with either 3% BSA or 5% milk, membranes were incubated overnight at 4 °C with rabbit antiphospho-Ser845-GluA1 (1:2000, Chemicon, Cat# AB5849; RRID:AB_92079), rabbit antiphospho-Ser-831-GluA1 (1:3000, Chemicon, Cat# AB5847; RRID:AB_92077), rabbit anti-GluA1 (1:4000, Millipore, Cat# AB1504; RRID:AB_2113602), rabbit anti-Kv4.2 (1:1000, Millipore, Cat# 07-491; RRID:AB_310662), rabbit anti-phospho-Ser438-Kv4.2 (1:100, Santa Cruz Biotech, Cat# sc-135551; RRID:AB_10839526), rabbit anti-phospho-Thr607-Kv4.2 (1:100, Santa Cruz Biotech, Cat# sc-22254-R; RRID:AB_2131823), mouse monoclonal anti-phospho-Thr602-Kv4.2 (1:2000, Santa Cruz Biotech, Cat# sc-16983-R; RRID:AB_670816), mouse monoclonal anti-phospho CaMKII Thr286 (Santa Cruz Biotech, Cat# sc-32289; RRID:AB_626786), mouse monoclonal anti- CaMKIIα (Santa Cruz Biotech, Cat# sc-13141; RRID: AB_626789) or rabbit anti-beta-actin (1:10,000, Proteintech, Cat# 60008-1; RRID:AB_2289225) primary antibodies. After washing, the membranes were incubated for 1 h with anti-rabbit or anti-mouse IgG secondary antibody both conjugated with horseradish peroxidase (HRP) (Proteintech) at room temperature. HRP activity was visualized by enhanced chemiluminescence with ECL Plus Western Blotting Detection System (GE Healthcare). Revelation was performed on photographic film for GluA1 experiments and using the LAS imager for Kv and CaMKII experiments. Band intensity was estimated using the Image J software. Beta-actin band density was employed as a loading control. The % phosphorylation for each AMPA GluA1 subunit target or Kv4.2 channel target was calculated normalizing the phosphorylated form band intensity by the band intensity of the total GluA1 or Kv4.2 immunostaining. Phosphorylation levels of Kv4.2 were difficult to detect in slice samples (but not whole tissue controls used for optimization, see Appendix A). In particular, results for phosphorylated Kv4.2 on Thr602 were not of the highest quality and although densitometric quantification has been performed it is not shown in this paper. Additional experiments were not considered since: (1) The obtained results do not suggest significant changes on visual inspection, suggesting additional experiments would not add more to the conclusions; (2) the antibodies for PKv4.2 were discontinued and (3) the increase in gel load necessary to perform those experiments would require a large increase in the number of animals, which in agreement with the 3R policy, was not considered sufficient to justify further experiments.

### 2.3. Materials

VIP (Novabiochem), [Ac-Tyr^1^, D-Phe^2^] GRF (1-29), PG 97-269, PG 99-465, Ro 25-1553 and [K^15^, R^16^, L^27^] VIP (1-7)/GRF (8-27) (all Phoenix peptides, Europe) were made up in 0.1 mM stock solution in CH_3_COOH 1% (v v-1). H-89 (N-[2-((p-bromocinnamyl)amino) ethyl]-5-isoquinolinesulfonamide) (Calbiochem), KN-62 (4-[(2S)-2-[(5-isoquinolinylsulfonyl) methylamino]-3-oxo-3-(4-phenyl-1-piperazinyl)propyl] phenyl isoquinolinesulfonic acid ester, Tocris Cookson, UK) and GF-109203X (3-[1-[3-(Dimethylamino)propyl]-1H-indol-3-yl]-4-(1H-indol-3-yl)-1H-pyrrole-2,5-dione) (Sigma/RBI) were made up in 5 mM stock solutions in DMSO. The maximal DMSO and CH_3_COOH concentrations used were devoid of effects on tritium release. Bicuculline methochloride (Ascent Scientific, UK), AP-5 (D-(-)-2-Amino-5-phosphonopentanoic acid), and SKF89976A (1-(4,4-Diphenyl-3-butenyl)-3-piperidinecarboxylic acid) (both Tocris Cookson, UK), were prepared in aqueous solution. Anti-VIP mouse monoclonal IgG1 (Santa Cruz, Cat# sc-57499; RRID: AB_630434) was obtained as a suspension in PBS and contained less than 0.1% NaN_3_ and 0.1% gelatine. The maximal concentrations of CH_3_COOH and NaN_3_ delivered to the slices, 0.001% (*v*/*v*) and 0.0004% (*p*/*v*), induced no change on fEPSP slope (*n* = 53). Stock solutions were kept frozen at −20 °C in aliquots until use. Aliquots were thawed and diluted in aCSF for use in each experiment.

### 2.4. Data and Statistical Analysis

LTP values are depicted as the mean ±S.E.M of *n* experiments. Each *n* represents a single experiment performed in slices obtained from one different animal for LTP experiments and a sample obtained from slices from 2–3 animals for western blot experiments. Statistical analysis was performed using GraphPad Prism 6.01 for Windows. The significance of the differences between the LTP means was evaluated using paired Student’s *t*-test. Repeated measures ANOVA with Tukey’s post-hoc test (when F was significant) was used to evaluate group differences in western blot experiments. Variance homogeneity was verified using Brown and Forsythe test. Normality of the data samples was confirmed using either the Shapiro-Wilk normality test or, for the smaller *n* samples, the Kolmogorov–Smirnov test. *p* < 0.05 was considered to represent statistically significant differences. No outliers were identified in our data (ROUT method).

## 3. Results

Representative fEPSP recorded under basal stimulation conditions in hippocampal slices from young-adult rats (Figure 1A) had an average slope of 0.647 ± 0.026 mV/ms (*n* = 33) that represented 40–60% of the maximal response in each slice. When mild TBS was applied to the control pathway (S1) an LTP was induced, causing a 28.9 ± 1.2% enhancement of fEPSP slope 50–60 min after TBS (*n* = 33, Figure 1A), an effect was absent (*n* = 4) when slices were stimulated in the presence of the NMDA receptor antagonist AP-5 (100 μM), as previously described. In the absence of added drugs, when a second TBS train was delivered to the test pathway (S2), an LTP of similar magnitude (% increase in fEPSP slope: 32.4 ± 3.6%, *n* = 4) to the one achieved in the control pathway (S1) was observed. When the selective VPAC receptor antagonist, Ac-Tyr^1^, D-Phe^2^ GRF (1–29) (300 nM) was delivered to the slices 20 min before S2, TBS stimulation elicited a larger LTP, now increasing by 48.6 ± 4.1% (*n* = 5) the fEPSP slope (Figure 1A,B). This suggests that endogenous VIP inhibits CA1 hippocampal LTP under these stimulation conditions and this urged us to test the involvement of each of the VIP-selective receptors in this action using the VPAC_1_-selective antagonist PG 97-269 [28] and the VPAC_2_-selective antagonist PG 99–465 [29]. The presence of PG 97-269 (100 nM) 20 min before TBS in S2 elicited a larger LTP, increasing by 51.3 ± 3.8% (*n* = 7) fEPSP slope (Figure 1C). Conversely, PG 99-465 (100 nM) present from 20 min before TBS in S2 resulted in an LTP of similar magnitude to the one obtained in control conditions (% increase in fEPSP slope: 34.6 ± 2.5%, *n* = 6, Figure 1D). Ac-Tyr^1^, D-Phe^2^ GRF (1-29) (300 nM), PG 97-269 (100 nM) or PG 99-465 (100 nM), when added to the slices, did not significantly alter fEPSP slope, suggesting that VIP does not carry out a tonic action on these receptors under basal stimulation conditions.

VPAC_1_ and VPAC_2_ receptors are activated by both VIP and PACAP endogenous ligands. To investigate if VIP was responsible for VPAC_1_ receptor activation leading to enhanced LTP, we tested whether an anti-VIP antibody could influence *mild TBS*-induced LTP. Delivery of mouse anti-VIP-IgG_1_ (0.4 μg/mL) to the slices did not significantly modify fEPSP slope under basal stimulation conditions, yet resulted in an enhanced LTP magnitude of 48.7 ± 2.9%, (*n* = 5, Figure 2A), hinting that VIP is the likely endogenous mediator of tonic VPAC_1_ receptor activation leading to inhibition of LTP. To further comprehend VIP actions, we also probed the impact of exogenously added VIP on LTP. When 1 nM and 10 nM VIP was added before S2 an enhancement of 20.4 ± 4.3% (*n* = 5) and 15.8 ± 3.4% (*n* = 5), respectively, in fEPSP slope was observed. TBS, applied when 1 nM VIP was present, caused a reduced LTP (% increase in fEPSP slope 14.6 ± 1.1%, *n* = 5, Figure 2B), whereas the presence of 10 nM VIP mildly enhanced LTP magnitude (% increase in fEPSP slope 43.7 ± 4.0%, *n* = 5, Figure 2C) compared to control conditions, suggesting that exogenously added VIP may be activating in a concentration dependent manner either different/additional receptors or distinct cellular and molecular pathways than endogenous VIP.

VIP-expressing interneurons either elicit a direct modulation of pyramidal cell excitability or regulate synaptic transmission to pyramidal cell dendrites by promoting disinhibition through two independent pathways [1]. To elucidate the role of GABAergic transmission in modulation of CA1 LTP by VPAC_1_ receptors, the impact of PG 97-269 on LTP was tested upon blockage of fast GABAergic transmission with bicuculline, a selective GABA_A_ receptor antagonist. When delivered to the slices before S1, bicuculline (10 μM) increased fEPSP slope by 41.5 ± 4.9% (*n* = 5). For this, stimulation was reduced to make fEPSP slopes of similar magnitude to the ones obtained in the lack of 10 μM bicuculline. TBS stimulation of control pathway (S1) in the presence of bicuculine (10 μM) caused an LTP of 25.1 ± 4.7% (*n* = 5, Figure 3A). When PG 97-269 (100 nM) was added to hippocampal slices in the presence of bicuculline, TBS in S2 elicited a similar LTP (% increase in fEPSP slope of 23.2 ± 4.8%, *n* = 5, Figure 3A), hinting that endogenous inhibition of TBS-induced LTP by VPAC_1_ receptors is reliant on modulation of GABAergic transmission.

VPAC_1_ modulation of GABA release in the hippocampus may target GABA exocytosis or depolarization-induced reversal of the nerve terminal GABA transporter 1 (GAT-1) [5]. To test the involvement of this pathway on modulation of hippocampal LTP by endogenous VPAC_1_ receptor activation, we tested the effect of PG 97-269 on LTP upon inhibition of GAT-1 transporters with the selective antagonist SKF89976a. TBS delivered to S1 in the presence of SKF89976a (5 μM) elicited an LTP, enhancing fEPSP slope by 33.8 ± 2.6% (*n* = 5, Figure 3B). When PG 97-269 (100 nM) was applied to hippocampal slices in the presence of SKF89976a, TBS in S2 stemmed a mildly larger LTP (% increase in fEPSP slope: 45.0 ± 2.8%, *n* = 5, Figure 3B), suggesting that VPAC_1_ receptor mediated inhibition of LTP by endogenous VIP is only partially reliant on the presynaptic control of synaptic GABA by the GAT-1 transporter.

We further investigated if VPAC_1_ mediated actions on LTP depended on the modulation of the TBS-induced NMDA-dependent component of LTP by testing the effect of PG 97-269 in the presence of the NMDA receptor antagonist (2R)-amino-5-phosphonopentanoate (AP-5). AP-5 (100 μM) did not change fEPSP slope under basal stimulation. TBS delivered to S1 in the presence of AP-5 (100 μM) abolished LTP (% change in fEPSP slope of −0.7 ± 3.2%, *n* = 4, Figure 4A). In the presence of PG 97-269 (100 nM), together with AP-5 (100 μM), TBS in S2 also did not cause any long-lasting change in fEPSP slope (−0.4 ± 1.9%, *n* = 4, Figure 4A), hinting a tonic VPAC_1_ receptor-mediated inhibition of NMDA receptor activation.

LTP expression is described to rely on the Ca^2+^-dependent activation and consequent auto-phosphorylation of Ca^2+^/calmodulin dependent protein kinase II (CaMKII), which in turn promotes the recruitment of AMPA GluA1 subunits [30]. Yet, this has also been reported as not essential for expression of NMDA-dependent LTP in the mouse [31]. The involvement of CaMKII in VPAC_1_ receptor-mediated enhancement of TBS-induced LTP by endogenous VIP was investigated using the CaMKII selective inhibitor KN-62. TBS stimulation of S1 in the presence of KN-62 (50 μM) suppressed LTP (% change in fEPSP slope: 1.0 ± 0.7% (*n* = 4, Figure 4B). TBS in S2 delivered in the presence of PG 97-269 (100 nM) and KN-62 (50 μM) elicited a very small LTP (% increase in fEPSP slope: 9.1 ± 1.3%, *n* = 4, Figure 4B). KN-62 (50 μM) did not significantly change fEPSP slope under basal stimulation conditions.

To further clarify the role of CaMKII in VPAC_1_ receptor-mediated modulation of CA1 LTP, we investigated the influence of the VPAC_1_ receptor antagonist on CaMKII phosphorylation induced by TBS. An enhancement of 25.2 ± 23.8% (*n* = 5, Figure 4C,D) in CamKII autophosphorylation in residue Thr286 50 min after TBS was observed. No significant changes in the total expression of CaMKIIα (*n* = 5, Figure 4C,D) were detected. The presence of the selective VPAC_1_ receptor antagonist PG 97-269 (100 nM) alone did not change total CaMKIIα or CamKII Thr286 autophosphorylation (*n* = 5, Figure 4C,D). When TBS was delivered in the presence of PG 97-269 (100 nM) the enhancement in CamKII Thr286 autophosphorylation was much higher (51.7 ± 30.0%, *n* = 5, Figure 4C,D) than the one observed in its absence but due to high variability of responses, this difference did not attain statistical significance (*p* > 0.05). Altogether, these observations suggest that tonic VPAC_1_ receptor-mediated inhibition of LTP is not fully dependent on CaMKII activity and involves additional transduction pathways.

To tackle this, the downstream targets of various intracellular kinases on AMPA GluA1 receptor subunits were examined as these were formerly identified as mediators of hippocampal LTP expression by promoting traffic or altering AMPA receptor opening probability [32]. As we previously described [26], 50 min after TBS we detected an enhancement of 50.2 ± 9.8% (*n* = 5, Figure 5C,D) in GluA1 phosphorylation in residue Ser831, targeted by both CaMKII and protein kinase C (PKC). No significant changes were observed in Ser845 phosphorylation, targeted by protein kinase A (PKA), or in the total expression of GluA1 subunits (*n* = 5, Figure 5A,B,E,F). The presence of the selective VPAC_1_ receptor antagonist PG 97-269 (100 nM) alone did not change total GluA1 or in GluA1 phosphorylation (*n* = 5, Figure 5A–F). When TBS was delivered in the presence of PG 97-269 (100 nM), the enhancement in GluA1 phosphorylation in residue Ser831 was not different (*p* > 0.05) than the one observed in the absence of the VPAC_1_ receptor antagonist (47.6 ± 14.3%, *n* = 5, Figure 5C,D) and phosphorylation on Ser845 was not altered. Altogether, this suggests that VPAC_1_ receptor activation by VIP during TBS LTP induction does not modify GluA1 phosphorylation and that the mechanism operated by VPAC_1_ receptors to regulate LTP induced by TBS lies elsewhere.

We further investigated the involvement of the G_s_/adenylate cyclase/PKA transduction system in VPAC_1_-mediated inhibition of CA1 LTP induced by TBS. In the presence of the selective PKA inhibitor H-89 (5 μM) [33], the effect of PG 97-269 (100 nM) on LTP induced by TBS was not changed (% increase in fEPSP slope: 44.9 ± 5.8%, *n* = 7, Figure 6A). In addition, we investigated the involvement of PKC on TBS-induced LTP. Upon selective inhibition of PKC with GF 109203X (1 μM) [34], the effect of PG 97-269 (100 nM) on LTP induced by TBS was similarly not changed (% increase in fEPSP slope: 44.2 ± 3.5%, *n* = 7, Figure 6B). When present during S1, H-89 (5 μM) and GF 109203X (1 μM) did not affect LTP (Figure 6A,B).

Finally, additional pathways involved in VPAC_1_ receptor modulation of hippocampal TBS-induced LTP were probed by investigating the targets of different intracellular kinases on Kv4.2 K^+^ channels, previously reported to contribute to LTP expression by suppressing the A-current and facilitating action potential backpropagation [35,36]. When Kv4.2 phosphorylation was examined 50 min after *mild TBS,* an enhancement of 167.8 ± 67.2% (*n* = 4, Figure 7A,B) was detected in Ser438 phosphorylation (a site targeted by CaMKII). The total expression of Kv4.2 subunits was also enhanced (*n* = 5, Figure 7E,F), yet no changes (*n* = 3–4) were observed in Kv4.2 phosphorylation in two ERK phosphorylation sites Thr607 (Figure 7C,D) and Thr602 (not shown in Figure 7). Basal phosphorylation (control conditions) at these two sites was low. When adding the selective VPAC_1_ receptor antagonist PG 97-269 (100 nM) in the absence of TBS, no significant changes in total Kv4.2 levels or in Kv4.2 phosphorylation were observed (*n* = 4, *p* > 0.05, Figure 7A–F). When TBS was delivered in the presence of PG 97-269 (100 nM), the enhancement in Kv4.2 phosphorylation in residue Ser438 was abolished (*n* = 4) and an enhancement of 61.1 ± 18.0%, in Thr607 phosphorylation (*n* = 3) was then observed. Altogether, this suggests that VPAC_1_ receptor activation by VIP during TBS LTP induction has a restraining effect on Thr607 phosphorylation, while facilitating Ser438 phosphorylation and thus preventing a large LTP from occurring.

## 4. Discussion

In the present work we describe for the first time that: (1) VIP, acting on VPAC_1_ (but not VPAC_2_) receptors, is an endogenous inhibitor of hippocampal LTP induced by mild TBS in young adult Wistar rats, an effect dependent on GABAergic transmission; (2) Tonic VPAC_1_-mediated inhibition of hippocampal LTP depends on of NMDA and CaMKII-dependent LTP expression mechanisms and is independent on PKA and PKC activity and (3) Inhibition of LTP by tonic VPAC_1_ activation enhances expression and Ser438 phosphorylation of Kv4.2 potassium channels and inhibits their Thr607 phosphorylation but does not depend on AMPA GluA1 phosphorylation.

The physiology of hippocampal VIP-expressing interneurons has been the focus of active investigation in recent years [14,37,38], yet the physiological role of endogenous VIP in the control of hippocampal circuits remains largely unexplored [1]. Endogenous VIP fine-tunes GABA release from hippocampal nerve terminals [5], hippocampal CA1 LTD [16] and is required for spatial discrimination during learning in water maze [39]. The progeny of VIP-deficient female mice show marked cognitive impairment in hippocampal-dependent tasks [15,40]. VIP neuropeptide, stored in large dense-core granules, is released by long-lasting depolarization or high-frequency neuronal firing [41,42], as likely occurs during TBS to allow its physiological control of CA1 LTP. Since our preliminary data suggested VIP was an endogenous modulator of hippocampal CA1 TBS-induced LTP [1], we investigated the cellular mechanisms and hippocampal receptors mediating these effects that may constitute important pharmacological targets to modulate hippocampal-dependent memory processes in pathological conditions like epilepsy, aging or neurodevelopmental disorders [43,44].

VIP acts through activation of two high-affinity receptors, VPAC_1_ and VPAC_2_ [1,5] and is expressed in CA1 area in at least three different interneuron populations, basket cells (BCs, Figure 8) which synapse mainly on pyramidal cell bodies and proximal dendrites, and several interneuron-specific (IS) interneuron populations [3,45]. We show that endogenous modulation of TBS-induced LTP by VIP depends on VPAC_1_ receptor activation and GABAergic transmission, suggesting a mechanism involving disinhibition. Since PACAP, that can also activate VPAC_1_ receptors, is present mostly in glutamatergic terminals, and can directly influence glutamatergic transmission, it is not likely the mediator of these effects.

VIP-expressing BCs in hippocampal CA1, about 8% of the VIP-immunoreactive hippocampal interneurons, are the main source of VIP that activates VPAC_2_ receptors, since VPAC_2_ immunoreactivity is mainly detected in the hippocampal *stratum*
*piramidale* [6,7] where most VIP-expressing BCs synapses reside [45], and does not influence synaptic transmission to pyramidal cell dendrites. These effects may also be endogenously mediated by PACAP, present in *stratum*
*piramidale* glutamatergic synapses [46]. Although VPAC_2_ receptor activation can enhance NMDA currents in CA1 pyramidal cells [17], it was not involved in the endogenous modulation of TBS-induced LTP reported in this work, suggesting BCs are not the cellular mediators of this effect. Alternatively, since VPAC_2_ receptors are expressed in lower amounts in the hippocampus, endogenous VIP release induced by TBS may not activate sufficient VPAC_2_ receptors to have a relevant effect on LTP expression.

VPAC_1_ receptors, mainly expressed in the *stratum*
*oriens* and *stratum*
*radiatum* of the Ammon’s Horn [6,7], can co-localize with glial markers, and mediate VIP effects of synaptic transmission to pyramidal cell dendrites. VPAC_1_ receptors promote disinhibition acting on pre- and postsynaptic components to modulate GABAergic transmission [1]. Although immunohistochemistry never demonstrated VPAC_1_ receptors in hippocampal interneurons, suggesting VPAC_1_-mediated actions could be due to glial regulation of synaptic GABA availability, VPAC_1_ receptor activation inhibits GABA release from hippocampal isolated nerve terminals [5], indicating the presynaptic presence of VPAC_1_ receptors in GABAergic synapses. In addition, the main glial cells regulating synaptic activity through the release of gliotransmitters [47] such as GABA, which could also promote disinhibition, are astrocytes. These cells express mostly VPAC_2_ receptors and few VPAC_1_ receptors [48], whereas VPAC_1_ receptors are expressed in microglia that mainly release gliotransmitters in pathological conditions [44,49]; thus, they are not relevant in physiological conditions such as TBS-like activity involved in synaptic plasticity. Altogether this supports our hypothesis of a neuronal target in the modulation of hippocampal LTP by VPAC_1_ mediated disinhibition.

VPAC_1_ receptor activation also mildly enhances NMDA currents in CA1 pyramidal cells [17], suggesting endogenous VIP can directly influence NMDA-dependent hippocampal LTP. Moreover, enhancement in VIPergic inputs to parvalbumin-expressing BCs in the CA3 is implicated in hippocampal-dependent spatial learning [50]. Yet, VPAC_1_ receptor-mediated modulation of CA1 NMDA receptor-dependent LTD [16] involves control of disinhibition, uncovering the importance of VIP-expressing IS interneurons in synaptic plasticity. In fact, disinhibition is crucial in the control of hippocampal synaptic plasticity [51] and synaptic inhibition at pyramidal cell dendrites is crucial for Ca^2+^-dependent input selectivity and precision of LTP induction [52]. The now described modulation of CA1 LTP by endogenous VIP is dependent on GABAergic transmission, implying that VIPergic modulation of hippocampal CA1 LTP targets disinhibition.

Several populations of CA1 VIP-expressing IS interneurons (Table 1, Figure 8) may initiate VIP regulation of disinhibition [3,45]: (1) Interneurons immunoreactive for VIP and calretinin, with cell bodies near the *stratum piramidale* and projecting to the *stratum-oriens-alveus* border (VIP^+^ CR^+^ O/A or IS3 interneurons); (2) VIP-expressing interneurons located at the *stratum radiatum-stratum piramidale* border (VIP^+^ CR^+/−^ SR) or (3) at the *stratum radiatum—Stratum lacunosum-moleculare* border (VIP^+^ IS2 interneurons) and projecting to the *stratum radiatum* [3,45] and (3) VIP long-range projecting (LRP) interneurons [53]. Their main synaptic inputs, neurochemical markers and target selectivity are summarized in Table 1 yet given their complex interconnections it is challenging to infer which generates the tonic actions on VPAC_1_ receptors inhibiting TBS-induced LTP. While VIP^+^ IS2 interneurons are mainly targeted by the temporoammonic (TA) pathway, and thus not likely recruited by Schaffer collateral (SC) electrical stimulation, the dendrites of IS3 and VIP^+^ CR^+/−^ SR interneurons span all layers and are likely activated by both SC and TA pathways yet IS3 interneurons receive a predominant inhibitory drive from both VIP^+^ IS2 and CR^+^ IS1 interneurons [38] and likely GABAergic projections from the medial septum, important in regulation of hippocampal theta activity [54,55]. Although VIP-expressing IS3 interneurons play a crucial role in goal-directed spatial learning tasks [14] and are mainly recruited during theta oscillations [38], their activation during TBS-induced LTP may have little impact under SC stimulation, since OLM interneurons control the somatic propagation of otherwise spatially confined TA inputs to pyramidal cells during theta activity [56].

VIP^+^ CR^+/−^ SR interneurons are considerably less studied than IS3 or IS2 cells, but the neurochemical profile of their synaptic targets suggests they innervate SR CB^+^ and CCK^+^ Schaffer collateral-associated (SCA) interneurons [14,45] involved in feedforward inhibition (FFI) to pyramidal cell dendrites. TBS-induced LTP relies on the suppression of phasic FFI by the first burst, which allows for enough depolarization and temporal summation to activate NMDA receptors [23,24,57] canonically attributed to inhibition of GABA release from feedforward interneurons by GABA_B_ autoreceptors [58]; yet, disinhibition may play an important role in this process. FFI to CA1 pyramidal cells is mediated by physiologically and morphologically distinct GABAergic interneurons: perisomatic-targeting BCs and dendritic-targeting bistratified cells and SCA cells [59,60]. Our results suggest that VIP^+^ CR^+/−^ SR interneurons provide maximal feedforward disinhibition (and VIP release) during TBS, unleashing LTP at proximal pyramidal cell dendrites, targeted by the SC pathway. VIP VPAC_1_ receptor-mediated inhibition of GABA release [5] at VIP^+^ CR^+/−^ SR to SCA interneuron synapses during theta activity would curtail disinhibition. Accordingly, upon VPAC_1_ receptor blockade, disinhibition would increase, eliciting the observed enhancement of TBS-induced LTP. Nevertheless, a few IS3 and VIP^+^ LRP interneurons target bistratified interneurons [38,53] and can thus elicit this same effect. In agreement, we showed that increasing synaptic GABA availability through partial inhibition of GAT-1 reduced the effect of VPAC_1_ receptor activation on TBS-induced LTP, suggesting that VPAC_1_ receptors prevent GABA spillover that could activate tonic instead of phasic inhibition, that shunts neuronal excitability, controls neuronal input-output gain and involves distinct extra-synaptic GABA_A_ receptors [61,62]. By controlling synaptic GABA levels [5], VIP acting on presynaptic VPAC_1_ receptors may provide timing precision to theta oscillation control by phasic inhibition.

In this work, tonic VPAC_1_ receptor activation during TBS-induced LTP did not significantly influence AMPA receptor GluA1 phosphorylation, either at Ser845 or Ser831, in agreement with previous observations showing that activation of hippocampal VPAC_2_ (but not VPAC_1_) receptors promote GluA1 phosphorylation at Ser845 [63], a PKA target site that is implicated in LTP maintenance and late-LTP [64].

Finally, we describe that tonic VPAC_1_ receptor activation during TBS-induced LTP facilitates Ser438 phosphorylation and Kv4.2 expression yet restrains Kv4.2 Thr607 phosphorylation, thus restraining LTP. Up-regulation of dendritic Kv4.2 mRNA by NMDA receptor activation [65] and TBS-induced enhancement of Kv4.2 phosphorylation in Ser438 following LTP in the hippocampus [26] have previously been reported, but the effect of VPAC_1_ receptor activation is utterly new.

Finally, the physiological relevance of this inhibitory control of LTP by endogenous VIP acting on VPAC_1_ receptors may well be related to neuroprotection against aberrant/excessive synaptic plasticity phenomena, which may in extreme cases contribute to epileptogenesis. In fact, our preliminary data suggests that blockade of VPAC_1_ receptors during in vitro epileptiform activity may protect against LTP saturation occurring in the following 1–2 h [43], a phenomena believed to contribute to seizure recurrence and aggravated damage. This may also impact cognitive deterioration in aging, Alzheimer’s disease or diseases associated with LTP/LTD imbalance like epilepsy or Down’s syndrome [43,66].

In conclusion, tonic VPAC_1_ receptor activation during TBS-induced LTP in the CA1 area of the hippocampus restrains LTP induction and expression by regulating inhibition of feedforward inhibitory cells. This brings into question the hypothesis that VIP VPAC_1_ receptor ligands, by fine-tuning disinhibition, may constitute more efficient and safer drugs to treat cognitive decline in the aging or epileptic.

## 5. Conclusions

This paper reports for the first time the endogenous modulation of TBS-induced LTP by endogenous activation of VPAC_1_ receptors. Since this effect is fully dependent on GABAergic transmission and is mimicked by application of a VIP-specific antibody, it is most likely mediated by VIP which is exclusively expressed in hippocampal interneurons. This is further co-substantiated by the fact that the transduction pathways mediating this effect were, for the most part, the same as the ones involved in TBS-triggered LTP induction and expression, such as activation of NMDA receptors and CaMKII activity. In addition, endogenous activation of VPAC_1_ receptors may indirectly influence the control of dendritic excitability by Kv4.2 channels. This suggests that inhibition of hippocampal LTP by VIP may be neuroprotective in conditions associated with aberrant synaptic plasticity, and that VPAC_1_ receptor ligands may constitute more efficient and safer drugs to treat cognitive decline in the aging or epileptic.

## Figures and Tables

**Figure 1 biology-11-00627-f001:**
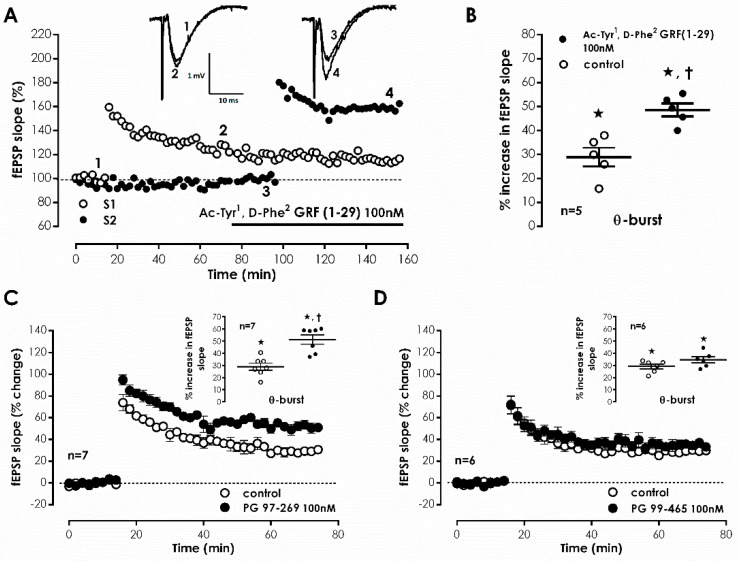
Endogenous VIP impairs hippocampal CA1 LTP through VPAC_1_ receptor activation. (**A**). Representative experiment of the time-course of changes in fEPSP slope elicited by theta-burst stimulation (TBS, 5 bursts at 5 Hz, each composed of four pulses at 100 Hz). In the same slice, the control pathway (–○–, S1) was stimulated in the absence of added drugs and the test pathway (–●–, S2) in the presence of the broad range VIP receptor antagonist Ac-Tyr^1^, D-Phe^2^ GRF (1-29) (100 nM). Inset: fEPSP traces obtained before (time points: 1 for S1 and 3 for S2) and 50–60 min after (time points: 2 for S1 and 4 for S2) TBS in the same experiment. Traces represent sequentially the stimulus artifact, the presynaptic volley and the fEPSP and are the average of eight consecutive responses. (**B**). LTP magnitude calculated as the averaged enhancement of fEPSP slope observed 50–60 min after TBS in the absence of added drugs (○) or in the presence (●) of Ac-Tyr^1^, D-Phe^2^ GRF (1-29) (100 nM). (**C**). and (**D**). Averaged time-course of fEPSP slope alterations caused by TBS in the absence of added drugs (–○–) or when either the selective VPAC_1_ antagonist PG 97-269 (100 nM, –●–, (**C**)) or the selective VPAC_2_ antagonist PG 99-465 (100 nM, –●–, (**D**)) were present. Inset: Magnitude of LTP calculated as the averaged enhancement of fEPSP slope observed 50–60 min after TBS without added drugs (○) or when PG 97-269 (100 nM, (**C**)) or PG 99-465 (100 nM, (**D**)) was added (●). Individual values and the mean ± S.E.M are shown (**B**–**D**). * *p* < 0.05 (Student’s *t*-test) as compared to the fEPSP slope before LTP induction; † *p* < 0.05 (paired Student’s *t*-test) as compared with absence of drugs in the same slices (○, in the left).

**Figure 2 biology-11-00627-f002:**
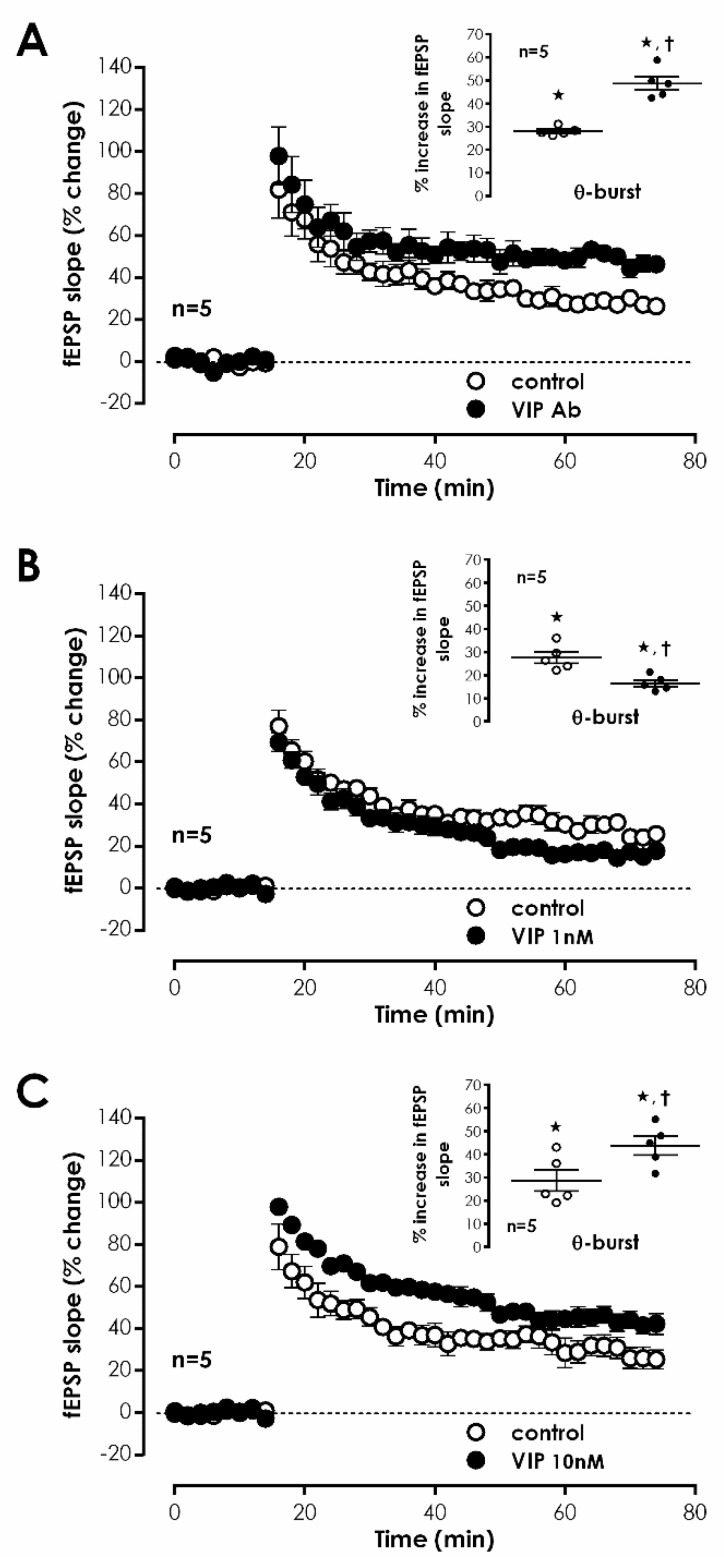
Tonic actions of VIP control hippocampal CA1 LTP in precise targets (**A**–**C**). Averaged time-course of alterations in fEPSP slope caused by theta-burst stimulation (TBS) in the absence of added drugs (–○–) and in the presence (–●–) of either a selective VIP antibody (0.4 µg/mL, (**A**)), VIP (1 nM, (**B**)) or VIP (10 nM, (**C**)). Control and test conditions (absence and presence of drugs) were assessed in independent pathways in the same slice. Inset: Magnitude of LTP calculated as the averaged enhancement of fEPSP slope observed 50–60 min after TBS obtained when no drugs were added (○) or when either a VIP antibody (0.4 µg/mL, (**A**)), or VIP in two different concentrations (1 nM, (**B**) and 10 nM, (**C**)) were present (●). Individual values and the mean ±S.E.M are represented. * *p* < 0.05 (Student’s *t*-test) as compared to the fEPSP slope before LTP induction; † *p* < 0.05 (paired Student’s *t*-test) as compared with absence of drugs in the same slices (○, in the left).

**Figure 3 biology-11-00627-f003:**
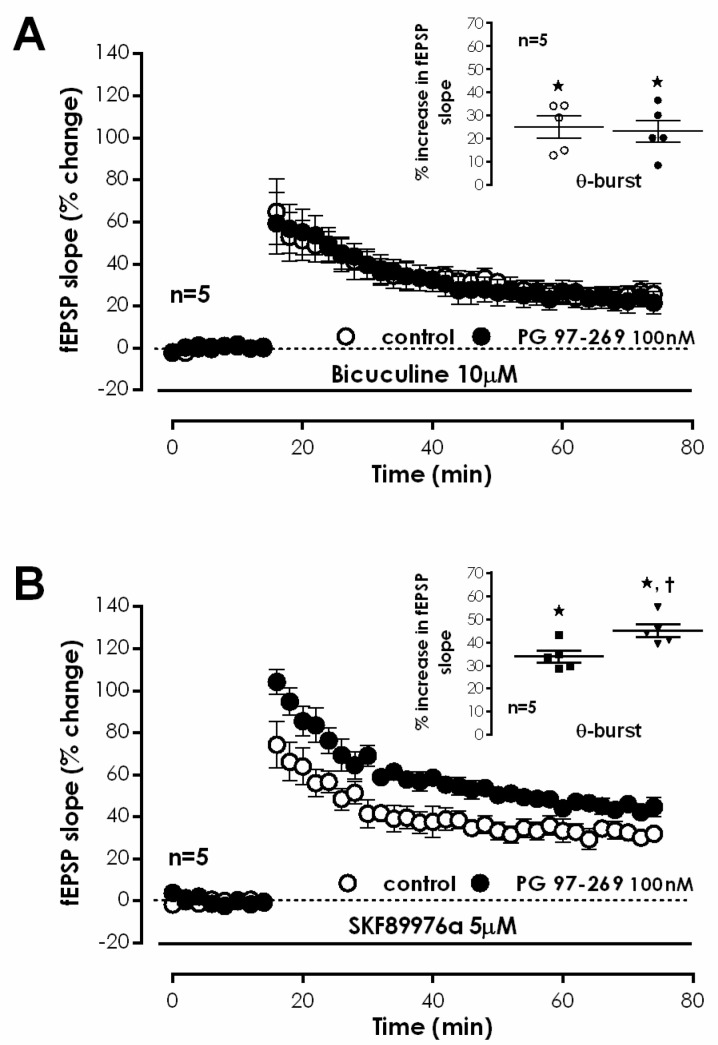
Endogenous VPAC1 receptor-mediated inhibition of hippocampal CA1 LTP is dependent on GABAergic transmission. (**A**,**B**) Averaged time-course alterations caused by theta-burst stimulation (TBS) in fEPSP slope when the selective VPAC_1_ antagonist PG 97-269 (100 nM) was absent (–○–) or present (–●–) and either the selective GABA_A_ receptor antagonist Bicuculline (10 μM, (**A**)) or the selective GAT_1_ GABA transporter inhibitor SKF8997A (5 μM, (**B**)) were added from the beginning of the experiment. Control and test conditions (lack and presence of PG 97-269) were assessed in two independent pathways of the same slice. Inset: LTP magnitude calculated from the averaged enhancement of fEPSP slope observed 50–60 min after TBS in the absence (○) or in the presence (●) the selective VPAC_1_ antagonist PG 97-269 (100 nM) when either Bicuculline (10 μM, (**A**)) or SKF8997A (5 μM, (**B**)) were present throughout the experiment. Individual values and the mean ± S.E.M are portrayed. * *p* < 0.05 (Student’s *t*-test) as compared to the fEPSP slope before LTP induction; † *p* < 0.05 (paired Student’s *t*-test) as compared with control conditions (absence of PG 97-269) in the same slices (○, in the left).

**Figure 4 biology-11-00627-f004:**
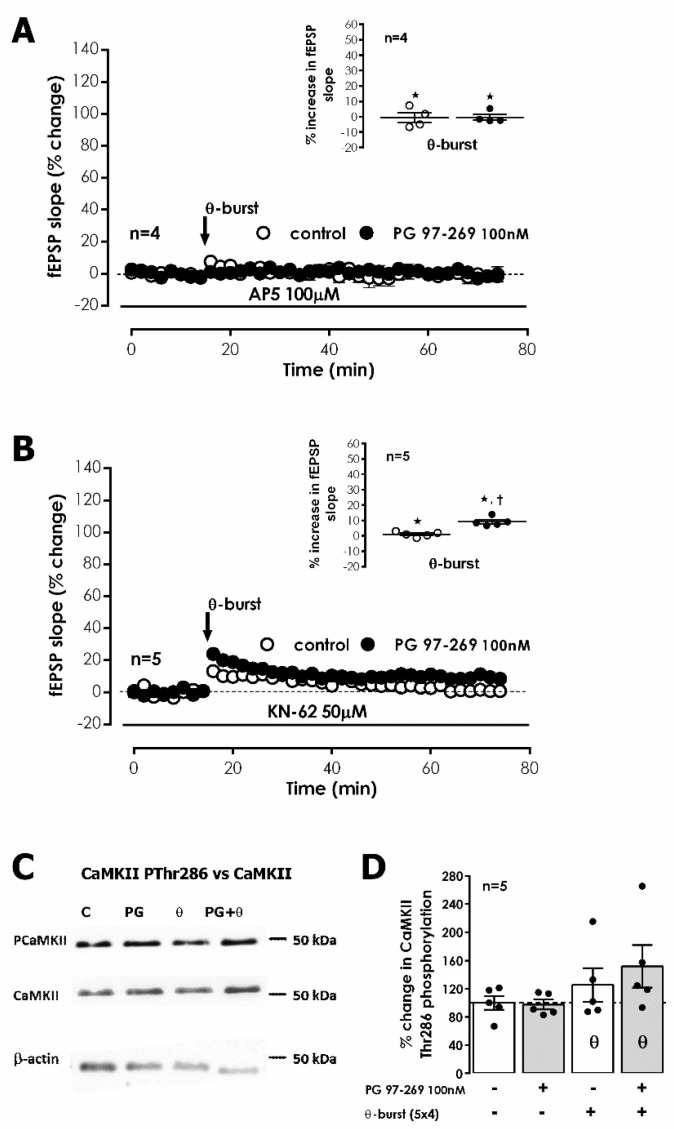
VIP VPAC1 receptor mediated endogenous inhibition of hippocampal CA1 LTP is dependent on NMDA activation and CaMKII activity. (**A**,**B**) Averaged time-course of alterations in fEPSP slope elicited by theta-burst stimulation (TBS) in the absence (–○–) and in the presence (–●–) of the selective VPAC_1_ antagonist PG 97-269 (100 nM) when either the selective NMDA receptor antagonist AP5 (100 μM, (**A**)) or the selective CaMKII inhibitor KN-62 (50 μM, (**B**)) were added from the beginning of the experiment. LTP elicited in the absence and presence of PG 97-269 were assessed in independent pathways of the same slice. Inset: LTP magnitude calculated as the averaged enhancement of fEPSP slope observed 50–60 min after TBS in the absence (○) or in the presence (●) of the selective VPAC_1_ antagonist PG 97-269 (100 nM) when either the selective NMDA receptor antagonist AP5 (100 μM, (**A**)) or the selective CaMKII inhibitor KN-62 (50 μM, (**B**)) were added from the beginning of the experiment. Individual values and the mean ± S.E.M are portrayed. * *p* < 0.05 (Student’s *t*-test) as compared to the fEPSP slope before LTP induction; † *p* < 0.05 (paired Student’s *t*-test) as compared with control conditions (absence of PG 97-269) in the same slices (○, in the left). (**C**,**D**). Western-blot immunodetection of CaMKII autophosphorylation in Thr286 and of total CaMKII expression obtained in one individual experiment where hippocampal slices were subjected to Schaffer collateral basal or theta-burst (θ) stimulation in the absence and in the presence (PG) of the selective VPAC_1_ antagonist PG 97-269 (100 nM). Slices were monitored for 50 min after theta-burst stimulation (or equivalent time for controls) before WB analysis. % CaMKII autophosphorylation in Thr286 residues is plotted (**D**). Individual values and the mean ± S.E.M of five independent experiments are depicted. 100%—averaged CaMKII autophosphorylation obtained in control conditions (absence of theta-burst stimulation and PG 97-269).

**Figure 5 biology-11-00627-f005:**
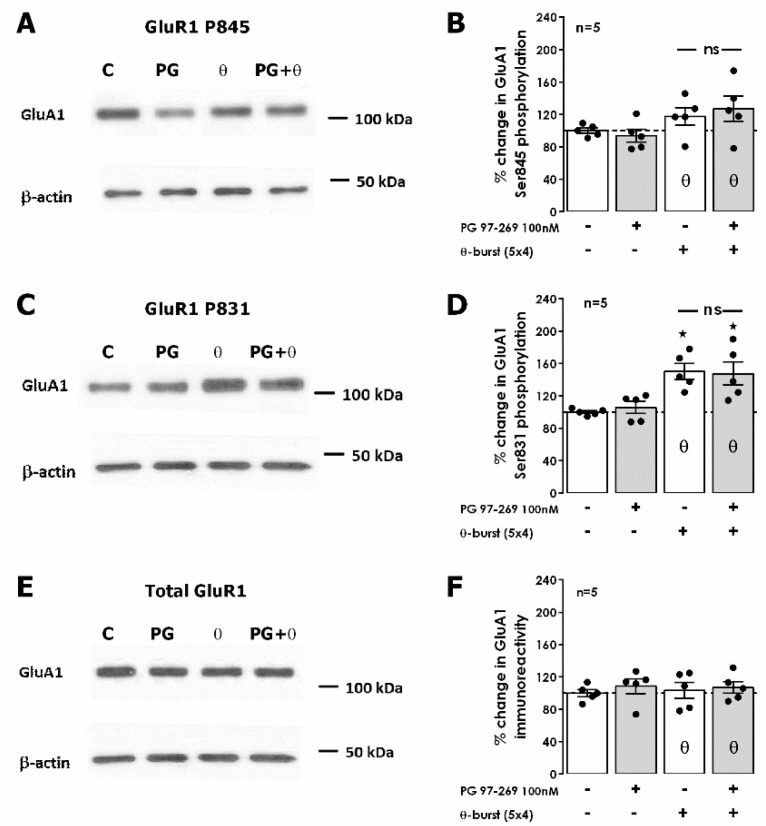
Impact of VPAC1 receptor inhibition and of theta-burst stimulation on phosphorylation of hippocampal AMPA GluA1 subunits on Ser845 and Ser831. (**A**,**C**,**E**). Western blot immunodetection of phosphorylated AMPA GluA1 in Ser845 and Ser831 and of total GluA1 subunits found in one experiment. Hippocampal slices were exposed to Schaffer collateral basal or theta-burst (θ) stimulation in the absence and in the presence (PG) of the selective VPAC_1_ antagonist PG 97-269 (100 nM). Slices continued basal stimulation for 50 min after TBS (or equivalent time for controls) before WB analysis. Total GluA1 immunoreactivity (**B**,**F**) % GluA1 phosphorylation in Ser845 (**D**) or Ser831 (**F**) residues are plotted. Individual values and the mean ± S.E.M of five independent experiments performed in duplicate are depicted. 100%—averaged GluA1 immunoreactivity or GluR1 phosphorylation obtained in control conditions (absence of theta-burst stimulation and PG 97-269). * *p* < 0.05 (ANOVA, Tukey’s multiple comparison test) as compared to lack of PG 97-269 and TBS; ns represents non-significant differences *p* > 0.05 (ANOVA, Tukey’s multiple comparison test) between respective bars.

**Figure 6 biology-11-00627-f006:**
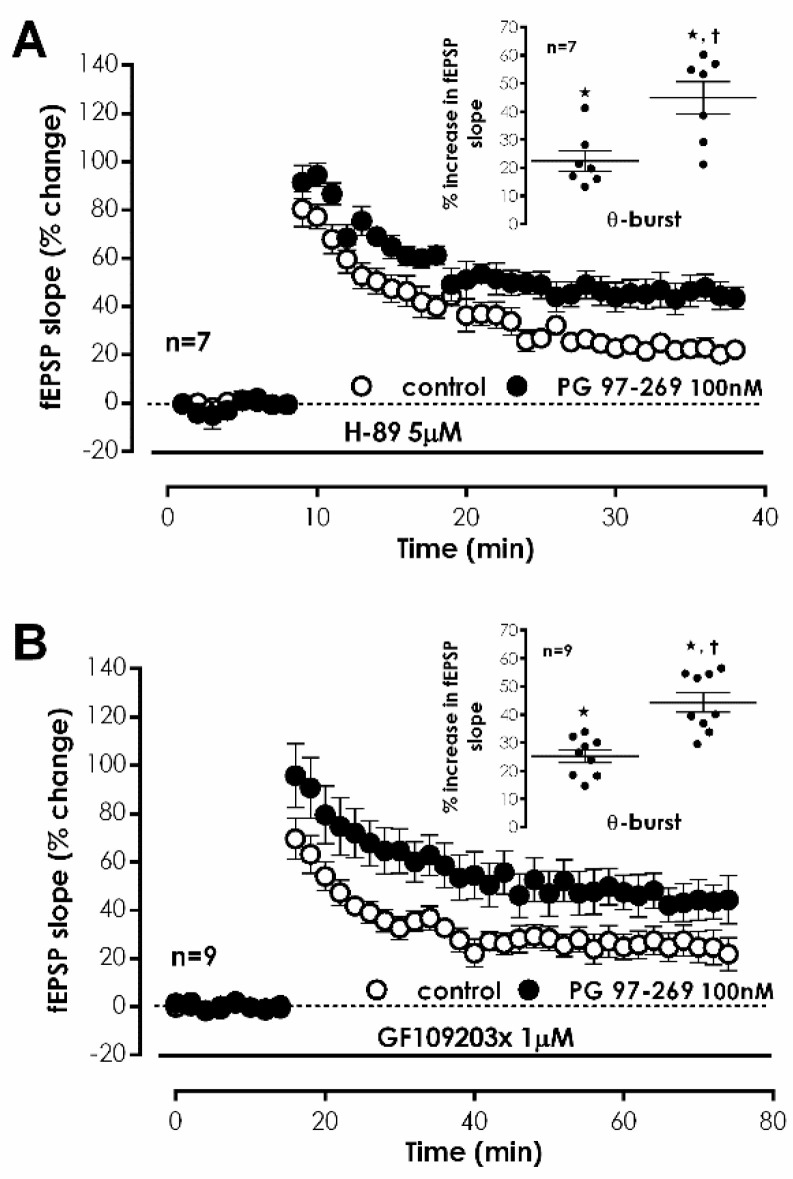
Endogenous inhibition of hippocampal CA1 LTP by VIP VPAC_1_ receptors does not depend on PKA and PKC activity. (**A**,**B**) Averaged time-course of changes in fEPSP slope caused by theta-burst stimulation in the absence (–○–) and in the presence (–●–) of the selective VPAC_1_ antagonist PG 97-269 (100 nM) when either the selective PKA inhibitor H-89 (5 μM, (**A**)) or the selective PKC inhibitor GF109203x (1 μM, (**B**)) were present from the beginning of the experiment. LTP in the absence and presence of PG 97-269 was elicited by TBS in independent pathways of the same slice. Inset: LTP magnitude calculated as the averaged enhancement of fEPSP slope observed 50–60 min after TBS in the absence (○) or in the presence (●) the selective VPAC_1_ antagonist PG 97-269 (100 nM) when either the selective PKA inhibitor H-89 (5 μM, (**A**)) or the selective PKC inhibitor GF109203x (1 μM, (**B**)) were present throughout the experiment. Individual values and the mean ± S.E.M are depicted. * *p* < 0.05 (Student’s *t*-test) as compared to the fEPSP slope before LTP induction; † *p* < 0.05 (paired Student’s *t*-test) as compared with control conditions (absence of PG 97-269) in the same slices (○, in the left).

**Figure 7 biology-11-00627-f007:**
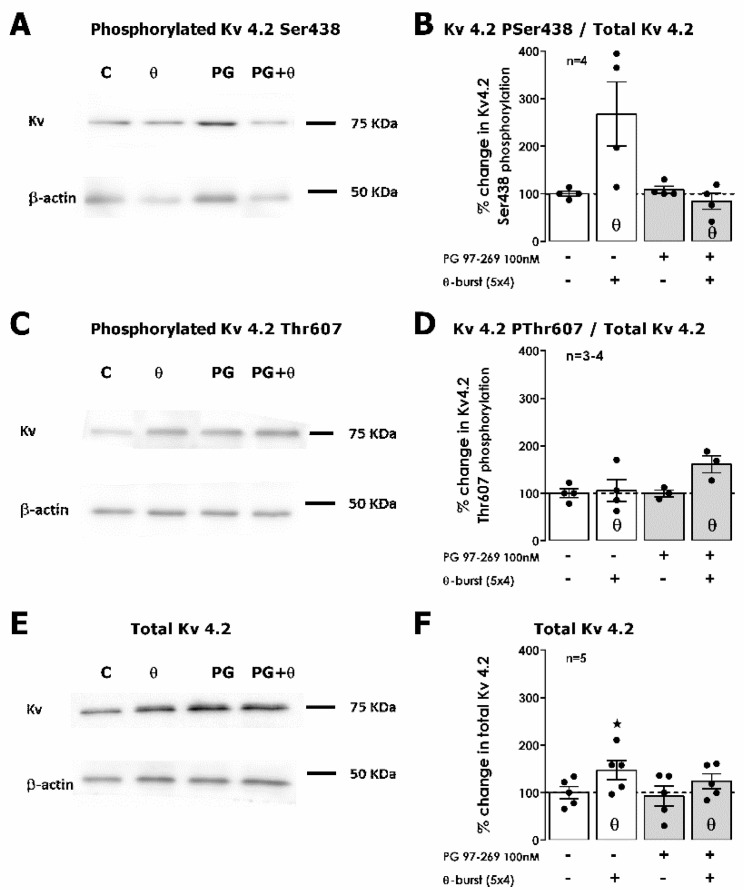
Impact of VPAC_1_ receptor inhibition on TBS-induced hippocampal Kv4.2 channel phosphorylation on Ser438, Thr802 and Thr807. Immunodetection of Kv4.2 phosphorylated forms in Ser438 (**A**), Thr607 (**C**) and total Kv4.2 (**E**) attained in one single experiment upon Schaffer collateral basal or theta-burst (θ) stimulation (TBS) of hippocampal slices in the absence or in the presence (PG) of the selective VPAC_1_ antagonist PG 97-269 (100 nM). Slices underwent basal stimulation for an additional 50 min after TBS (or equivalent time for controls) before WB analysis. Kv4.2 phosphorylation in Ser438 (**B**) or Thr607 (**D**) residues were normalized to the total variation in Kv4.2 (**F**) immunoreactivity. Individual values and the mean ± S.E.M of 3–4 independent experiments performed in duplicate are depicted. 100%—averaged Kv4.2 immunoreactivity obtained in control conditions (absence of TBS for either no added drugs or the presence of the VPAC_1_ antagonist PG 97-269 100 nM). * Denotes *p* < 0.05 (Student’s *t*-test) as compared to absence of TBS.

**Figure 8 biology-11-00627-f008:**
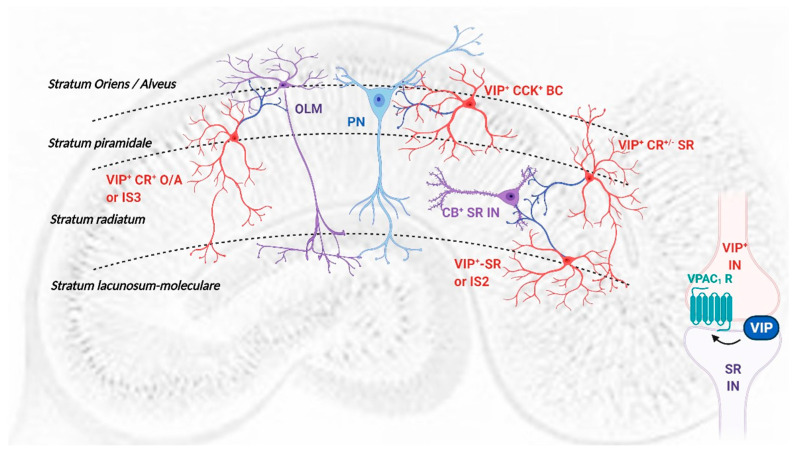
Representation of local VIP-expressing interneurons in the CA1 area of the rat hippocampus: layer location and target selectivity. PN: pyramidal neuron (light blue); Interneurons (purple); VIP-containing interneurons (red) and their axonal projections (dark blue); OLM—*Stratum*
*oriens* interneuron projecting to the *Stratum*
*lacunosum-moleculare*; CB^+^ SR IN—*Stratum*
*radiatum* calbindin 28K-expressing local interneurons. *VIP^+^-CCK^+^ BCs*: VIP and CCK-expressing *basket cells*; *VIP^+^ CR^+^ O/A*: VIP and calretinin-containing interneuron-selective interneuron targeting the *stratum*
*oriens*/*Alveus* or IS3 cells; *VIP^+^ SR*: *stratum*
*radiatum**/**lacunosum-moleculare* VIP-containing interneuron-selective interneuron targeting the *stratum*
*radiatum* or IS2 cells and *VIP^+^ CR^+/−^ SR: stratum*
*radiatum**/**piramidale* VIP-containing interneuron-selective interneuron targeting the *stratum*
*radiatum* (density of dendritic arborization is not represented; see Table 1 for details on main inputs).

**Table 1 biology-11-00627-t001:** Neuroanatomical features and connectivity of local-bound VIP-expressing interneurons in the CA1 area of the hippocampus. BCs—basket cells; CB—calbindin 28K; CCK—cholecystokinin; CR—calretinin; EC—Entorhinal cortex; Ins—interneurons; IS—interneuron-specific; LRP—long-range projection; OLM—*Oriens-Lacunosum moleculare* (interneurons); SC—Schaffer-collateral; SP—*Stratum*
*piramidale*; SR—*stratum*
*radiatum*; VIP—vasoactive intestinal peptide(X^+^);—expressing; (X^−^)—non-expressing [3,14,37,38,45].

	Dendritic Tree	Soma	Inputs	Projections	Target Cells in CA1
**VIP^+^ CCK^+^ BCs**	*Stratum Piramidale*	*Stratum Piramidale*	Schaffer collateral /commissural fibres	*Stratum Piramidale* and near *radiatum*	Pyramidal neurons
**VIP^+^ CR^+^ IS2 INs**	*Stratum lacunosum-moleculare*	*Stratum* *Radiatum/lacunosum-moleculare border*	Temporoammonic EC fibres	*Stratum radiatum*	VIP^+^ CCK^+^ BCs VIP^+^ CR^+^IS3 INs CB^+^ SC-associated INs
**VIP^+^ CR^+^ IS3 INs**	All layers	*Stratum Piramidale/radiatum border*	Schaffer collateral /commissural fibres Temporoammonic EC fibres IS1 and IS2 INs	*Stratum Oriens & Alveus*	Mostly OLM INs but also other *Oriens**/Alveus* INs projecting to the *SLM.* BCs Bistratified cells
**VIP^+^ CR^+/−^ SR/SP INs**	All layers	*Stratum Piramidale/radiatum border*	Schaffer collateral /commissural fibres	*Stratum radiatum and piramidale*	BCs Bistratified cells CB^+^ SC-associated INs
**VIP** **^+^ LRP INs**	*Stratum Oriens*	*Stratum Oriens*	Schaffer collateral/commissural fibres	*Stratum Oriens/Piramidale* *Subiculum*	Mostly OLM INs BCs Bistratified cells Pyramidal cells and INs

## Data Availability

The data that support the findings of this study are available from the corresponding author upon reasonable request. Western blot data available are depicted in Appendix A. Some data may not be made available because of privacy or ethical restrictions.

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
