# Peer review of "Endogenous VIP VPAC_1_ Receptor Activation Modulates Hippocampal Theta Burst Induced LTP: Transduction Pathways and GABAergic Mechanisms"

_biology, 2022, doi:10.3390/biology11050627_

Round 1

Reviewer 1 Report

In this in vitro study,  authors using hippocampus slices (CA1), isolated from young-adult  male  Wistar  rats, induced  long- term potentiation (LTP) by  Theta -burst stimulation (TBS) in the areas of the slices. Field excitatory post -synaptic potential (fEPSP) was recorded extracellularly.  Using non-specific and specific vasoactive intestinal peptide (VIP) receptors (VCAP1 and VCAP2) antagonists aimed to show the role of endogenous VIP on LTP.To display the mediators responsible for the VIP action on LTP, GABAA receptor and GAT1 antagonist were added to the medium. Authors using western blood technique, evaluated the potential pathways molecules have role in affecting LTP such as NMDA and CaMKII-dependent LTP, Ser438 phosphorylation of Kv4.2 potassium channels, AMPA GluA1 phosphorylation, PKA, and PKC activity. 

The authors concluded that endogenous VIP decreases the hippocampal CA1 LTP. VIP acting on VPAC1 (but not VPAC2) receptors, acts as an endogenous inhibitor of hippocampal LTP which is induced by mild TBS in young adult Wistar rats. The effect of VIP is dependent on GABAergic transmission.

They have also tested the effect of exogen VIP (1nM and 10 nM) and, found that lower dose of VIP (1nM) decreases LTP. On the other hand, the higher dose of VIP (10nM) mildly enhanced LTP magnitude (Fig 2C). Finally, a conclusion was made in abstract, lines 28-30: “ Altogether this suggests that endogenous VIP controls the expression of hippocampal CA1 LTP by regulating disinhibition through activation of VPAC1 receptors in interneurons. This may impact the autophosphorylation of CaMKII 28 during LTP, as well as the expression and phosphorylation of Kv4.2 K+ channels at hippocampal pyramidal cell dendrites. And page 22 line 528-31:” In conclusion, tonic VPAC1 receptor activation during TBS-induced LTP in the CA1 area of the hippocampus restrains LTP induction and expression by regulating inhibition of feedforward inhibitory cells. This brings into question the hypothesis VIP VPAC1 receptor ligands, by fine-tuning disinhibition, could constitute efficient and safer drugs to treat cognitive decline in aging or epilepsy.”

However, the manuscript is not acceptable in this form and needs revision.

The aim of the study should explain more clearly, more experiments should add to prove the suggestions of the authors, results of the study should be discussed and impacted learning and memory effect of VIP: What is the main importance of endogenous VIP modulates the LTP in the hippocampal special area? Discussion part of the manuscript needs carefully revision, it should rewritten easily understandable manner, and should concentrate their  main findings. Critics could help with the revisions.

The aim of the study is not clearly explained that why  hippocampal LTP is directed to VIP? In addition, although authors have not been measuring VIP concentration in the tissue medium, how decided that endogenous VIP is responsible for reducing LTP? Furthermore, examining exogen VIP in two different doses (1nM and 10 nM) was found that a lower dose of VIP (1nM) decreased LTP, higher dose (10nM) mildly enhanced LTP magnitude (Fig 2C). In that case, why similar molecular mechanisms such as receptors antagonism, channels activity, and protein kinase pathways were not tested for exogen VIP to compare the data of suggested endogenous VIP? There has no idea for the concentration of endogenous VIP released in the tissue medium. Is it low or high? There is no information that TBS stimulation causes VIP release. If the stimulation induced VIP release the quantitative amount of VIP could have been measured.

As for the receptors, it is well known that the VPAC1 receptors are constitutive receptor of VIP. On the other hand, VPAC2receptors are inducible receptors, which means that activation of the receptors is depends on changed circumstances. For this reason, VPAC2 receptors could show lower expression in the tissues such as the hippocampus. Therefore, in the discussion, for LTP comments, keep in mind that applied TBS may not sufficient to activate VPAC2 receptors.

GABA is the main inhibitory neurotransmitter in the central nervous system,  it involves in complex circuits throughout the central nervous system, it has a very significant effect on various physiologic performances. GABA releases either neurons or Astrocytes cells. GABA released from astrocytes has crucial effect on neuronal functions. Astrocytes can transform glutamatergic excitation to GABAergic inhibition.  Although authors are not tested the source of GABA, at least they should discuss it in the MS that VPAC1 mediated GABAergic disinhibitory circuits partly could associate with astrocytes.

In addition, PACAP also binds VPAC1 receptor  similar affinity with VIP. Hence, whether the observations on VIP-mediated LTP function reflected VIP or PACAP signaling on VPAC1 receptors remains unclear. The authors should clarify this situation.  If there is no reasonable explanation to eliminate this obscurity, PACAP antagonist should test by adding in the slices’ medium.

LTP a long-lasting strengthening of the response of a postsynaptic nerve cell to stimulation across the synapse that occurs with repeated stimulation and is thought to be related to learning and long-term memory. LTP involves the strenghtening of synapses. According to the results, what are the comments of the authors about the effect of VIP (endogenous) on the synaptic build-up and memory storage? Is it good or bad?

MS focused on the effect of VIP on GABA transmission modulation resulting in reduced LTP. There are some in vivo studies reported that VIP increases GABA level in the special areas of the brain and could have a beneficial effect for  Parkinson’s disease and Epilepsy. Those of the following articles should cite in the MS.

Vasoactive intestinal peptide (VIP) treatment of Parkinsonian rats increases thalamic gamma-aminobutyric acid (GABA) levels and alters the release of nerve growth factor (NGF …

OT Korkmaz, N Tunçel, M Tunçel, EM Öncü, V Şahintürk, M Çelik

Journal of molecular neuroscience 41 (2), 278-287

Vasoactive intestinal peptide (VIP) conducts the neuronal activity during absence seizures: GABA seems to be the main mediator of VIP

OT Korkmaz, S Arkan, EM Öncü-Kaya, N Ateş, N Tunçel

Neuroscience Letters 765, 136268

2021

Modulation of corpus striatal neurochemistry by astrocytes and vasoactive intestinal peptide (VIP) in parkinsonian rats

İH Yelkenli, E Ulupinar, OT Korkmaz, E Şener, G Kuş, Z Filiz, N Tunçel

Journal of molecular neuroscience 59 (2), 280-289

Page 3 line 74… Authors declared that ” A preliminary account of some of the results has been published as an abstract.” Journal (for abstract) and  Congress  should remark in the MS ( Journal of Molecular Neuroscience   January 2010 and  The 9th International Symposium on VIP, PACAP, and Related Peptides, October 5–8, 2009,Kagoshima, Japan. Abstract of the authors is given at the end of the critics.

If their preliminary data was published in JOMN 2010, the authors should explain some exactly similar experiments whether repeated or not (data are very close).  Additionally, their previously found contrary data for PKC is involving the LTP modulation, also needs explanation in the discussion part of the recently submitted MS. (Page 18 line 422: “Tonic VPAC1-mediated inhibition of hippocampal LTP  depends on of NMDA and CaMKII-dependent LTP expression mechanisms and is independent on PKA and  PKC activity and 3).”

Page 5 line 122: PPF abbreviation? Is it Paired Pulse Facilitation?

Previous abstract of the authors:

Journal of Molecular Neuroscience   January 2010 and  The 9th International Symposium on VIP, PACAP, and Related Peptides, October 5–8, 2009,Kagoshima, Japan. 

On the cellular and molecular pathways involved in VIP inhibition of LTP in the CA1 area of the hippocampus.Journal of Molecular Neuroscience 42(3):278

Vasoactive intestinal peptide (VIP) modulates hippocampal synaptic transmission through several receptor and cellular mechanisms (Cunha-Reis et al., 2004; 2005; 2006) and is fully dependent on GABAergic transmission. VIP containing interneurones are innervated by septal GABAergic and median raphe serotonergic fibres (Papp et al., 1999), suggesting an involvement in theta-related synaptic plasticity. We now evaluated how endogenous VIP influences hippocampal long-term potentiation (LTP) induced by theta-burst stimulation and what are the receptor and transduction pathways involved in this modulation. The role of VIP modulation GABAergic transmission was also investigated. Extracellular electrophysiological recordings in hippocampal slices were used to access LTP induced by theta-burst stimulation (5 x 100Hz, 4 stimuli, separated by 200 ms) Selective VPAC1 (PG 97-269) and VPAC2 (PG 99-465) as well a non selective (Ac-Tyr1 GRF(1-29)) VIP receptor antagonists were used to evaluate the involvement of endogenous VIP in hippocampal synaptic plasticity. The involvement of protein kinases A (PKA) and C (PKC) in these effects was studied using the selective inhibitors H-89 and GF109203x, respectively. How changes in GABAergic transmission contribute to this effect of VIP was tested in the presence of the selective GABAA antagonist biccuculline. Theta-burst stimulation caused an enhancement of 28±2.8% (n=15) in fEPSPs slope recorded 50-60 min after stimulation. Ac-Tyr1 GRF(1-29) (100nM) increased theta burst-induced LTP to 49±4.1% (n=4). PG 97-269 (100nM) also increased that theta burst-induced LTP to 40±6.7% (n=6). PG 99-465 (100nM) did not significantly change theta-burst induced LTP. Inhibition of PKA with H-89 (1mM, n=4) did not significantly change the enhancement caused by PG 97-269 on LTP but that effect was abolished upon inhibition of PKC with GF109203x (1mM, n=3). Blockade of GABAA receptors with bicuculline (10mM) also abolished VPAC1 receptor modulation of LTP induced by theta-burst stimulation (n=5). These results suggest that endogenous VIP has a restraining effect on hippocampal LTP through tonic activation of VPAC1 receptors and PKC. This effect is also dependent on GABAergic transmission suggesting an indirect effect of VIP on hippocampal glutamatergic synapses, involving modulation of hippocampal GABAergic circuits. Supported by FCT. Cunha-Reis D et al. (2004) Br J Pharmacol 143:733. Cunha-Reis D et al. (2005) Brain Res 1049:52. Cunha-Reis D et al. (2006) Ann NY Acad Sci 1070: 210. Papp EC et al. (1999) Neuroscience 90:369.

Author Response

The authors wish to thank the reviewer’s comments, which contributed to improvement of the manuscript. Regarding the specific points raised please find bellow the author’s response.

  1. The aim of the study should explain more clearly, more experiments should add to prove the suggestions of the authors, results of the study should be discussed and impacted learning and memory effect of VIP:

What is the main importance of endogenous VIP modulates the LTP in the hippocampal special area?

Discussion part of the manuscript needs carefully revision, it should rewritten easily understandable manner, and should concentrate their main findings. Critics could help with the revisions.

R: Our main findings are described in the first paragraph of discussion. Discussion was improved with several suggestions from the three reviewers, namely by highlighting the importance endogenous VIP may play in restraining LTP and avoiding its pathological saturation (page 17).

  1. The aim of the study is not clearly explained that why hippocampal LTP is directed to VIP?

R: As already mentioned in the text in the second paragraph of discussion, we studied the influence of endogenous VIP on LTP because neuropeptides are usually released by repetitive firing like the one occurring during TBS stimulation, and in addition VIP is an important modulator of hippocampal-dependent cognition, thus suggesting that endogenous VIP may be a modulator of hippocampal LTP. Furthermore, we found previously that under basal stimulation VPAC1 antagonists do not influence synaptic transmission, suggesting endogenous VIP release does not occur under those conditions. This is now also mentioned in the introduction (last paragraph, page 4, highlighted in yellow).

  1. In addition, although authors have not been measuring VIP concentration in the tissue medium, how decided that endogenous VIP is responsible for reducing LTP?

R: The use of an antibody specific for VIP (see Fig. 2.A) was used to demonstrate the similar effect of endogenous VIP removal and VPAC1 antagonists on LTP. We do not think it is feasible to dose synaptic VIP in a superfused hippocampal slice since the released neuropeptide is either not readily accessible, because being very brief during TBS it will not extravasate the synapse, or, when leaving the synaptic environment, it will be shortly washed away.

  1. Furthermore, examining exogen VIP in two different doses (1nM and 10 nM) was found that a lower dose of VIP (1nM) decreased LTP, higher dose (10nM) mildly enhanced LTP magnitude (Fig 2C). In that case, why similar molecular mechanisms such as receptors antagonism, channels activity, and protein kinase pathways were not tested for exogen VIP to compare the data of suggested endogenous VIP?

R: Our main goal in this study was to demonstrate the effect of endogenous VIP and receptor/transduction mechanisms on modulation of LTP. Since several endogenous cellular sources for VIP are present in the hippocampal slice, this would be the best approach to avoid misinterpretation of the actions of exogenous VIP or VPAC1 agonists, since not all these cell sources may be recruited during TBS. By applying exogenous VIP, we would activate also VPAC1 and VPAC2 receptors in multiple cellular targets (including massive activation of microglia and astrocytes) that may not participate in endogenous modulation of LTP by VIP, given the brief and mild intensity stimulation provided by TBS. In fact, we believe that studying the transduction pathways involved in LTP modulation by exogenous VIP would not bring clarity to this question. In addition, higher exogenous VIP concentrations may already activate PAC1 receptors, that bind PACAP with much higher affinity than VIP, and that have previously been reported to modulate LTP.

  1. There has no idea for the concentration of endogenous VIP released in the tissue medium. Is it low or high? There is no information that TBS stimulation causes VIP release. If the stimulation induced VIP release the quantitative amount of VIP could have been measured.

R: Estimating the exact concentration of VIP released in a superfused slice, as is often done with microdialysis in vivo, is difficult to achieve, since, as mentioned above, it is not easy to estimate how much it is immediately washed away. Furthermore, what is in fact relevant for modulation of LTP is synaptically released VIP, and not VIP spilled over the extracellular medium. In this respect, estimating VIP in a slice could in fact lead to erroneous conclusions. In addition, sensitivity of methods used so far to estimate VIP release from slices was only enough to evaluate VIP release un much more extreme conditions (e.g., veratridine-induced depolarization).

  1. As for the receptors, it is well known that the VPAC1 receptors are constitutive receptor of VIP. On the other hand, VPAC2receptors are inducible receptors, which means that activation of the receptors is depends on changed circumstances. For this reason, VPAC2 receptors could show lower expression in the tissues such as the hippocampus. Therefore, in the discussion, for LTP comments, keep in mind that applied TBS may not sufficient to activate VPAC2 receptors.

R: In the discussion, p15 (highlighted in yellow), a comment on the fact that lower levels of VPAC2 receptors may render their activation by TBS-triggered endogenous VIP release negligible to observe an effect on LTP was added. However, please bear in mind that VPAC2 receptors are mostly expressed in the pyramidal cell layer, likely modulating pyramidal cell excitability, while we are studying in this paper the synaptic plasticity at pyramidal cell dendrites.

  1. GABA is the main inhibitory neurotransmitter in the central nervous system, it involves in complex circuits throughout the central nervous system, it has a very significant effect on various physiologic performances. GABA releases either neurons or Astrocytes cells. GABA released from astrocytes has crucial effect on neuronal functions. Astrocytes can transform glutamatergic excitation to GABAergic inhibition. Although authors are not tested the source of GABA, at least they should discuss it in the MS that VPAC1 mediated GABAergic disinhibitory circuits partly could associate with astrocytes.

R: Given that hippocampal VIP is exclusively expressed by GABAergic interneurons, that VPAC1 receptor-mediated control of GABA release was previously demonstrated by our group, and that glial VPAC1 receptors were found mostly in microglia and only at very low levels in astrocytes, the main source of synaptic gliotransmitters that have been reported to mediate disinhibition and LTP in the hippocampus, it is not likely that astrocytes are the source of disinhibition triggered by VPAC1 receptors. Gliotransmitters can also originate from activated microglia in pathological conditions, but not likely in this physiological-mimicking scenario. This is discussed in the text on page 15 (highlighted in yellow).

  1. In addition, PACAP also binds VPAC1 receptor similar affinity with VIP. Hence, whether the observations on VIP-mediated LTP function reflected VIP or PACAP signalling on VPAC1 receptors remains unclear. The authors should clarify this situation. If there is no reasonable explanation to eliminate this obscurity, PACAP antagonist should test by adding in the slices’ medium.

R: In fact, an antagonist would not solve this question since antagonists are selective for the receptors and not for the ligands. The only antagonist that would block only the actions of PACAP is the one blocking PAC1 receptors, that do not bind VIP with high affinity. As discussed above, the use of a VIP specific antibody was used to demonstrate the involvement of VIP and not PACAP in this effect. Furthermore, PACAP, unlike VIP, is not involved in the modulation of GABAergic transmission in the hippocampus, since it is released from glutamatergic nerve terminals, and exerts a direct effect on pyramidal cells. Thus, the dependency of GABAergic transmission observed for the effect of VPAC1 receptors on LTP also suggests that it is VIP, and not PACAP, the endogenous mediator of this effect. This is discussed on page 14, third paragraph, highlighted in yellow.

  1. LTP a long-lasting strengthening of the response of a postsynaptic nerve cell to stimulation across the synapse that occurs with repeated stimulation and is thought to be related to learning and long-term memory. LTP involves the strengthening of synapses. According to the results, what are the comments of the authors about the effect of VIP (endogenous) on the synaptic build-up and memory storage? Is it good or bad?

R: As discussed in the text, endogenous VIP restrains LTP induced by TBS. The relevance of these effects may be related to neuroprotection in states of hyperexcitability. So, VIP does not contribute to synaptic build up and probably does not promote memory storage. This is not necessarily bad, since we want our synaptic build up to be precise and not dysregulated by abnormal hyperexcitable states. This is now discussed at the end of the discussion (highlighted in yellow).

  1. MS focused on the effect of VIP on GABA transmission modulation resulting in reduced LTP. There are some in vivo studies reported that VIP increases GABA level in the special areas of the brain and could have a beneficial effect for Parkinson’s disease and Epilepsy. Those of the following articles should cite in the MS.

Vasoactive intestinal peptide (VIP) treatment of Parkinsonian rats increases thalamic gamma-aminobutyric acid (GABA) levels and alters the release of nerve growth factor (NGF …

OT Korkmaz, N Tunçel, M Tunçel, EM Öncü, V Şahintürk, M Çelik

Journal of molecular neuroscience 41 (2), 278-287

Vasoactive intestinal peptide (VIP) conducts the neuronal activity during absence seizures: GABA seems to be the main mediator of VIP

OT Korkmaz, S Arkan, EM Öncü-Kaya, N Ateş, N Tunçel

Neuroscience Letters 765, 136268

2021

Modulation of corpus striatal neurochemistry by astrocytes and vasoactive intestinal peptide (VIP) in parkinsonian rats

İH Yelkenli, E Ulupinar, OT Korkmaz, E Şener, G Kuş, Z Filiz, N Tunçel

Journal of molecular neuroscience 59 (2), 280-289

R: The findings in these papers are now considered in several points of the discussion (highlighted in yellow).

  1. Page 3 line 74… Authors declared that ” A preliminary account of some of the results has been published as an abstract.” Journal (for abstract) and  Congress  should remark in the MS ( Journal of Molecular Neuroscience   January 2010 and  The 9th International Symposium on VIP, PACAP, and Related Peptides, October 5–8, 2009,Kagoshima, Japan. Abstract of the authors is given at the end of the critics.

If their preliminary data was published in JOMN 2010, the authors should explain some exactly similar experiments whether repeated or not (data are very close). Additionally, their previously found contrary data for PKC is involving the LTP modulation, also needs explanation in the discussion part of the recently submitted MS. (Page 18 line 422: “Tonic VPAC1-mediated inhibition of hippocampal LTP  depends on of NMDA and CaMKII-dependent LTP expression mechanisms and is independent on PKA and  PKC activity and 3).”

R: Regarding the results on influence of PKC on VPAC1 action on LTP in the abstract, those were preliminary results (n=3) that we could not manage to reproduce. The results of the following experiments were as described in this paper, i. e., the effect of the VPAC1 antagonist was not significantly altered by the PKC inhibitor. This is in fact the reason that the number of replicates is higher for these experiments (n=7), and two different batches of PKC inhibitor were used. The expression of TBS-induced LTP is itself is, in our observations, independent on PKC activity. This would not preclude a PKC-dependent modulation of LTP, that has previously been described in the literature.

  1. Page 5 line 122: PPF abbreviation? Is it Paired Pulse Facilitation?

R: Yes, abbreviation was now replaced by full text.

Reviewer 2 Report

The manuscript entitled " Endogenous VIP VPAC1 receptor activation modulates theta-burst induced LTP in the hippocampus: transduction pathways and GABAergic mechanisms" by Caulino-Rocha et al. investigate the modulation of CA1 LTP induced by theta-burst stimulation (TBS) by endogenous VIP release in hippocampal slices from young-adult Wistar rats using selective VPAC1 and VPAC2 receptor antagonists in vitro. For this purpose, electrophysiological recording and Western-blot techniques were used.

This study is original since no reports are available in the literature. The study is extensive, the data are robust, and adds to the body of knowledge on VIP and its receptors roles and physiology on the central nervous system. Therefore, deserves to be published, but there are only some minor issues that should be fixed to increase the quality of the research:

- In the method section, the placement of electrodes for stimulation on the Schaffer collateral fibers on hippocampal slides is mentioned. It would also be better to mention how such a small spot can be accurately aimed with electrodes so that the reader can understand. Is it distinguishable under the microscope? A special histological technique is being determined? Can it be precisely targeted through a stereoteximectic apparatus? How is repeatability?

- There are two studies, possibly by the same group, that show that VIP reduces absence seizures via GABAergic neurons in WAG/Rij rats (https://doi.org/10.1016/j.neulet.2021.136268) and prevents atrophy of related brain regions in Alzheimer's model transgenic mice (https://doi.org/10.1007/s12031-018-1226-8). It will be more informative for the reader to refer to these studies, both in terms of supporting the hypothesis of this study and discussing the effects of VIPergic neurons and their receptors in hippocampal neurons. 

Author Response

The authors wish to thank the reviewer’s comments, which contributed to improvement of the manuscript. Regarding the specific points raised please find bellow the author’s response.

  1. In the method section, the placement of electrodes for stimulation on the Schaffer collateral fibres on hippocampal slides is mentioned. It would also be better to mention how such a small spot can be accurately aimed with electrodes so that the reader can understand. Is it distinguishable under the microscope? A special histological technique is being determined? Can it be precisely targeted through a stereoteximectic apparatus? How is repeatability?

R: On page 6, the visual inspection that allows us to stimulate in the right place is now described. In the wrong place we do not get a response, or we get a signal with a different shape. The experiments were performed in vitro so no stereotaxic methods can be applied. Reproducibility is very good and the technique is used since the 70s.

  1. There are two studies, possibly by the same group, that show that VIP reduces absence seizures via GABAergic neurons in WAG/Rij rats (https://doi.org/10.1016/j.neulet.2021.136268) and prevents atrophy of related brain regions in Alzheimer's model transgenic mice (https://doi.org/10.1007/s12031-018-1226-8). It will be more informative for the reader to refer to these studies, both in terms of supporting the hypothesis of this study and discussing the effects of VIPergic neurons and their receptors in hippocampal neurons.

R: These references were now considered at several points of the discussion (highlighted in yellow or green).

Reviewer 3 Report

  1. Full names of abbreviations should be shown at first appearance in the manuscript. E.g. “LTP”
  2. The quality of western blotting figures is not high. E.g., Fig. 4 C and Fig 7A, actin didn’t show an equal loading.

Author Response

The authors wish to thank the reviewer’s comments, which contributed to improvement of the manuscript. Regarding the specific points raised please find bellow the author’s response.

  1. Full names of abbreviations should be shown at first appearance in the manuscript. E.g. “LTP”

R: The term LTP is now defined in the abstract.

The quality of western blotting figures is not high. E.g., Fig. 4 C and Fig 7A, actin didn’t show an equal loading.

R: We usually work with hippocampal membrane suspensions rather than solubilized proteins, to avoid extensive denaturation of membrane proteins for a long time before WB. We usually get better signals form membrane proteins in this way. Such membrane suspensions can be harder to handle because they tend to easily deposit with gravity within minutes. As such, this can be an inconvenience for reproducible protein loading and is happening in some of our experiments in a reasonable amount. All results are however in acceptable range for densitometric quantification and are normalised to loading control to account for these differences. Results that do not meet these standards are either excluded or not considered for quantitative evaluation. This is detailed at the end of WB description in the methods section.
